# Compensation Method for Polarization Mixing in the Homodyne Interferometer

**Chaoqun Wang** [1,2]**, Qiangxian Huang** [1,*]**, Xuemeng Ding** [1]**, Rongjun Cheng** [1]**, Liansheng Zhang** [1]**, Ruijun Li** [1] **and Hongli Li** [1]

[1] School of Instrument Science and Opto-Electronics Engineering, Hefei University of Technology, Hefei 230009, China; wangchaoqun@hfut.edu.cn (C.W.); dingxm674538134@163.com (X.D.); chengrj@hfut.edu.cn (R.C.); lszhang@hfut.edu.cn (L.Z.); rj-li@hfut.edu.cn (R.L.); hlli@hfut.edu.cn (H.L.)
[2] School of Electronic Science and Applied Physics, Hefei University of Technology, Hefei 230009, China
* Correspondence: huangqx@hfut.edu.cn; Tel.: +86-150-5510-1156

**Abstract:** A homodyne interferometer is one of the most important tools in nanometre measurements. However, its nonlinear error seriously affects measurement accuracy at the sub-nanometre level. As one of the dominant factors that cause nonlinear error in a homodyne interferometer with a quadrature detector system, the imperfection of polarizing beam splitters (PBSs) is investigated in this paper. The nonlinear error caused by the imperfection of PBSs in the detection part can be reduced by adjusting the gains of detectors. Nevertheless, eliminating the nonlinear error caused by the polarization mixing of the PBS in the interferometer part is difficult. In this paper, the nonlinear error caused by the polarization mixing of the PBS in the interferometer part is analyzed, and an optical compensation method is proposed to correct this polarization mixing. Theoretical calculation and simulation analysis show that this method can reduce the effect of inherent polarization mixing on nonlinear error significantly. In comparison with using only gain adjustment, the nonlinear error can be reduced by two orders of magnitude when the proposed method is applied. The nonlinear error can be decreased from approximately 4.5 nm to approximately 0.045 nm using the presented method based on the simulation results.

**Keywords:** homodyne interferometer; nonlinearity; polarization mixing; compensation

## 1. Introduction

The requirements for dimensional measurement accuracy are increasing with the rapid development of nanotechnology. A homodyne interferometer has become one of the most important tools for nanometre measurement because of its high measurement accuracy and traceability to international length standards [1]. However, the measurement accuracy of a homodyne interferometer with a quadrature detector system is restricted by nonlinear errors [2].

Several compensation methods for reducing the nonlinearity have been suggested. An ellipse-fitting method was proposed firstly by Heydemann [3], which was later improved by Brich and Wu [4,5]. In this method, two raw signals are fitted into a general elliptical equation using the least-squares method and then the nonlinear coefficients are calibrated from the calculated parameters of the general elliptical equation. Therefore, the total nonlinear errors including the interferometer itself and the signal detection system can be corrected, thereby proving that this approach is a comprehensive method. However, this method is time-consuming and unsuitable for real-time compensation because the coefficients of the elliptical equation should be calculated prior to the real measurement, which requires considerable computational effort. Then, Hu devised a fine correction method involving Heydemann correction [6]. The proposed method suppresses the nonlinearity in a homodyne interferometer to

0.18 nm, which would be 1.49 nm with conventional Heydemann correction. However, the step-by-step fine-ellipse fitting is quite time-consuming and is not suitable for real-time measurement. Köning further studied the Heydemann correction and proposed a new ellipsefitting method based on the minimization of the geometric distance between the measured and fitted signal values [7,8]. The statistical uncertainty of the Heydemann correction is analyzed in detail, which has a high dependency on the signal-to-noise ratio of signals.

Methods using hardware-based correction have been proposed to eliminate the three primary errors (DC offsets, unequal AC amplitudes and quadrature phase error). A real-time correction technique based on the use of a microcontroller and computational analogue circuits has been developed, and the better the quality of the quadrature signals, the more accurate the method is [9]. Meanwhile, Hu provides a real-time compensation of the variable cyclic errors utilizing a field-programmable gate array in a homodyne laser interferometer [10].

Others have proposed their solutions to improve the optical configurations. Yan proposed an optical structure that consists of two homodyne interferometers in which two orthogonal single-frequency beams share a common reference arm and partial measurement arm [11,12]. A series of experiments demonstrate that the proposed interferometer is able to realize sub-nanometer accuracy without nonlinearity for nanometer displacement and nanometer accuracy for millimeter displacement. Cui has proposed an optical layout based on a non-polarizing beam splitter (NPBS) and balanced interference between two circularly polarized beams [13]. Besides the replacing polarizing beam splitters (PBSs) of the detection part with Wollaston prisms (WPs) with a high extinction ratio ($1 \times 10^5$), the PBS in the interference part is replaced with an NPBS, the half-wave plate (HWP) and the quarter-wave plate (QWP) in the detection part are removed. The optical layout is simple, but the cost of the WP is more than the cost of the PBS. For a homodyne interferometer, existing research indicates that the quadrature phase error is the main effect that produces nonlinearity error, and the polarization mixing introduced by the poor performance of the PBSs is the main error source that causes the quadrature phase error [13–16]. The nonlinearity caused by the imperfections of the PBSs has been studied by some researchers, and several compensation methods have been proposed. Keem has proposed a method of adjusting the gains of the electrical detection circuit to reduce the nonlinear error caused by the PBSs [17,18]. Although this simple and noteworthy method can remove the offset and different amplitudes induced by the PBSs in the detection part, it cannot eliminate the polarization mixing caused by the PBS in the interferometer part. Ahn has proposed a passive compensation method by realigning the axes of wave plates at specific angles according to the characteristics of the PBSs, instead of the conventional angles of wave plates [19]. The difficulty of this method lies in the high demand for an accurate test to determine the performance of the PBSs prior to setting up the interferometer. In addition, gain and offset correction should be performed on raw interference signals before the angles of the wave plates are optimized. Hu recently designed an enhanced homodyne interferometer without DC offsets [16]. Two PBSs in the detection part are replaced with WPs with a high extinction ratio, and then, the gain and phase correction methods are combined to reduce the nonlinear error. However, the compensation for the polarization mixing of the PBS in the interferometer part remains lacking.

Although several studies have been conducted to reduce the nonlinearity of a homodyne interferometer, the nonlinear error caused by the imperfection of the PBS in the interferometer part has not yet been compensated. The primary reason is the difficulty in eliminating the polarization mixing caused by the PBS in the interferometer part using gain correction or other methods. In the current study, an optical layout that can compensate for the polarization mixing caused by the PBS in the interferometer part is proposed. By combining it with the gain-adjustment method, it can reduce the nonlinear error by two orders of magnitude compared with using only gain correction.

## 2. Nonlinearity in a Homodyne Interferometer

A typical homodyne interferometer is shown in Figure 1. The interferometer has a quadrature detector system, and it can be divided into an interferometer part and a detection part. In general,

one PBS is located in the interferometer part, whereas two PBSs are in the detection part. A rectangular coordinate system is established, where the propagation direction of the beam is along the Z-axis. The X- and Y-axes are parallel and perpendicular to the paper surface, respectively. The beam from the polarizer P has two orthogonally polarized components: the p-polarized and s-polarized components. This beam is split into two polarized beams by the $PBS_1$ in the interferometer part, which are then are reflected separately by the reference mirror $M_r$ and the measuring mirror $M_m$. Consequently, a phase difference $\varphi$, which is dependent of the displacement of the measuring mirror, is induced between the two beams. The two polarized beams are combined at the end of the interferometer part. The polarization states of the measuring and reference beams are rotated by 45° and 135°, respectively, and become 45° and −45° after the combined beam passes through the HWP in the detection part. Then, the intensity of the combined beam is evenly split into two beams by the NPBS, and the two beams are split again by the $PBS_2$ and the $PBS_3$ in the detection part into four beams to generate interference signals $I_1$, $I_2$, $I_3$ and $I_4$, with a 90° phase difference between each.

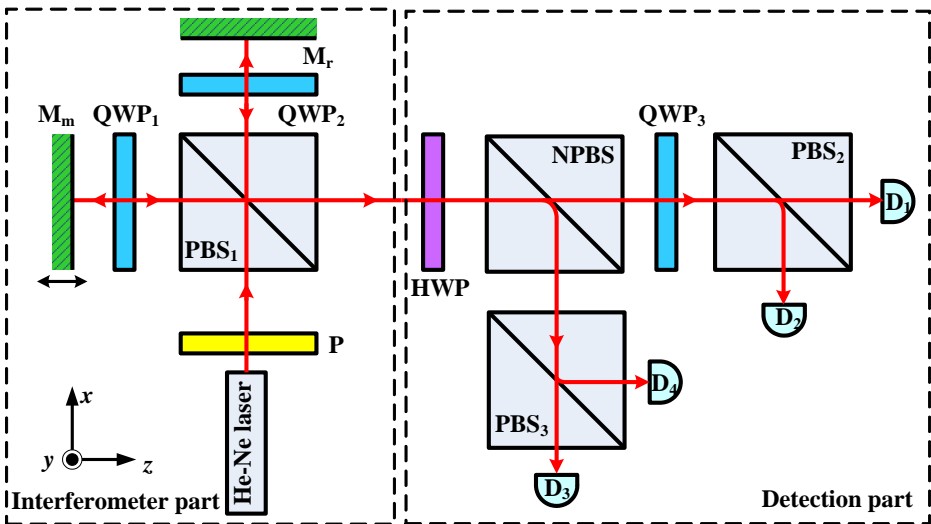

**Figure 1.** Homodyne interferometer with a quadrature detector system. P is the polarizer; $PBS_1$, $PBS_2$ and $PBS_3$ are the polarizing beam splitters; $M_r$ and $M_m$ are the mirrors; $QWP_1$, $QWP_2$ and $QWP_3$ are the quarter-wave plates; HWP is the half-wave plate; NPBS is the non-polarizing beam splitter; and $D_1$, $D_2$, $D_3$ and $D_4$ are the photoelectric detectors.

The intensity of the four beams is detected by four photoelectric detectors. Two phase-quadrature signals $I_{0x}$ and $I_{0y}$ can be obtained from the four signals. Ideally, the phase-quadrature signals $I_{0x}$ and $I_{0y}$ should be two sinusoids with a 90° phase difference. Therefore, the relative phase difference $\varphi$ with respect to the displacement of the measuring mirror can be derived using the following equations:

$$I_{0y} = A\sin(\varphi) \qquad I_{0x} = A\cos(\varphi)$$
$$\varphi = \arctan\left(\frac{I_{0y}}{I_{0x}}\right) \tag{1}$$

where $A$ is the amplitude of the ideal phase-quadrature signals.

The homodyne interferometer is a relative displacement measurement system. $L_0$ represents the relative displacement of the measuring mirror, which is directly proportional to the relative phase difference $\varphi$ between the reference and measuring beams (assuming that the refractive index of the atmosphere is 1). This relationship can be ideally expressed as:

$$L_0 = \frac{\lambda}{4\pi}\varphi \tag{2}$$

where $\lambda$ is the wavelength of the laser.

Nevertheless, ensuring that the phase difference between the two quadrature signals is equal to 90° is difficult due to the imperfections of the optical components. The relationship between the real displacement $L$ and the phase difference $\varphi$ of the interference intensity can be rewritten as follows:

$$L = \frac{\lambda}{4\pi}\left[\arctan\left(\frac{I_y}{I_x}\right) + m\pi\right]$$
$$m = 0, 1, 2, \ldots$$
$$(3)$$

where $I_x$ and $I_y$ are the quadrature signals in a real case. Therefore, the nonlinear error $\Delta L$ of the displacement can be expressed as follows:

$$\Delta L = \frac{\lambda}{4\pi}\left\{\phi - arc\tan\left(\frac{I_y}{I_x}\right) - m\pi\right\}$$
$$m = 0, 1, 2, \ldots$$
$$(4)$$

There are many optical components in the homodyne interferometer with quadrature detection system, which have not only manufacturing errors but also setup errors associated with their angular positioning. All of these form the error sources of nonlinearity errors. For the pair of quadrature signals in Equation (3), the nonlinear errors are mainly embodied in three forms: DC offsets, unequal AC amplitudes and quadrature phase error. Some error sources induce DC offsets of the quadrature signals, others produce unequal amplitudes of the quadrature signals, and some cause quadrature phase error of the quadrature signals. Most optical components have no effect on quadrature phase error. For example, the misalignment of the fast axes and phase retardation error of the wave plate only induce DC offsets and unequal AC amplitudes, which can be easily corrected with hardware circuits or software correction algorithms, but not quadrature phase error, which is more difficult to correct. The polarization mixing of the poor performance of the PBSs is the main source that introduces quadrature phase error. We focus on correction of the polarization mixing of the imperfection of the PBSs, so all of the optical components are assumed to be ideal here, except for the PBSs.

Here, the Jones matrix and the Jones vector are used to describe an optical element and a polarized beam, respectively. Suppose that the optical components except for the PBSs are ideal. Then, the angles of the polarized beams are exact. In an ideal case, the Jones matrices of the wave plates, PBS, NPBS and incident beam can be expressed as follows:

$$
\begin{aligned}
Q &= \frac{\sqrt{2}}{2}\begin{bmatrix} 1 & -i \\ -i & 1 \end{bmatrix} & H &= \frac{\sqrt{2}}{2}\begin{bmatrix} 1 & 1 \\ 1 & -1 \end{bmatrix} \\
T_0 &= \begin{bmatrix} 1 & 0 \\ 0 & 0 \end{bmatrix} & R_0 &= \begin{bmatrix} 0 & 0 \\ 0 & -1 \end{bmatrix} \\
T_n &= \frac{\sqrt{2}}{2}\begin{bmatrix} 1 & 0 \\ 0 & 1 \end{bmatrix} & R_n &= \frac{\sqrt{2}}{2}\begin{bmatrix} 1 & 0 \\ 0 & -1 \end{bmatrix} \\
E_0 &= \begin{bmatrix} 1 \\ 1 \end{bmatrix}E_0
\end{aligned}
$$
$$(5)$$

where $Q$ and $H$ are the ideal Jones matrices of the QWP and HWP, respectively. $T_0$ and $R_0$ are the transmittance and the reflectance matrices of PBS, respectively. $T_n$ and $R_n$ are the transmittance and the reflectance matrices of NPBS, respectively. $E_0$ is the electric field vector of the incident beam, the polarization axis of which is at 45°.

In a real case, the PBSs exhibit different optical characteristics with respect to p- and s-polarized beams. On the one hand, a leakage of an unwanted polarized component exists, which causes the polarization mixing. Therefore, a small part of the s-polarized component can pass through the p-polarized beam direction, and vice versa. On the other hand, a slight optical loss is induced by the absorption of the PBS. That is, the transmissivity-to-reflectivity ratio of the PBS is not equal to 1.

To describe the imperfect beam-splitting phenomenon of a PBS, the Jones matrix can be modified as follows:

$$T = \begin{bmatrix} t_p & 0 \\ 0 & t_s \end{bmatrix} \qquad R = \begin{bmatrix} r_p & 0 \\ 0 & r_s \end{bmatrix} \tag{6}$$

where $T$ and $R$ represent the modified matrices of the transmittance and reflectance of the PBS, respectively. In addition, $t_p{}^2$ and $t_s{}^2$ denote the transmitted intensity of the p- and s-polarized beams, respectively, whereas $r_p{}^2$ and $r_s{}^2$ represent the reflected intensity of the p- and s-polarized beams, respectively. Under ideal conditions, $t_p{}^2$ and $r_s{}^2$ are equal to 1, whereas $t_s{}^2$ and $r_p{}^2$ are equal to zero. For an actual PBS, however, $t_p{}^2$ and $r_s{}^2$ are less than 1, whereas $t_s{}^2$ and $r_p{}^2$ are not equal to zero.

The electric field vectors of the reference beam $E_r$ and the measuring beam $E_m$ at the end of the interferometer part can be written as follows:

$$\begin{aligned} E_r &= R_1 \cdot Q_2 \cdot Q_2 \cdot T_1 \cdot E_0 \cdot e^{i\varphi_r} \\ E_m &= T_1 \cdot Q_1 \cdot Q_1 \cdot R_1 \cdot E_0 \cdot e^{i\varphi_m} \end{aligned} \tag{7}$$

where $\varphi_m$ and $\varphi_r$ are the phases that correspond to the optical paths of the reference and measuring mirrors, respectively. The vectors of the four interference beams at the end of the detection part can be written as a group of Jones matrices as follows:

$$\begin{aligned} E_1 &= T_2 \cdot Q_3 \cdot T_n \cdot H \cdot (E_r + E_m) \\ E_2 &= R_2 \cdot Q_3 \cdot T_n \cdot H \cdot (E_r + E_m) \\ E_3 &= T_3 \cdot R_n \cdot H \cdot (E_r + E_m) \\ E_4 &= R_3 \cdot R_n \cdot H \cdot (E_r + E_m) \end{aligned} \tag{8}$$

The subscript $i$ ($i$ = 1, 2, 3) in $T_i$, $R_i$ and $Q_i$ indicates the number of the PBSs and the QWPs, respectively. $E_j$ is the electric field vector of the four beams that reach the photoelectric detector, where the subscript $j$ ($j$ = 1, 2, 3, 4) indicates the number of photoelectric detectors. The intensity signals can be obtained through a preamplifier with an adjustable gain $k_j$, where $j$ takes values of 1 to 4. Then, the two phase-quadrature signals $I_x$ and $I_y$ that can be obtained by subtracting the value of one detector from that of another detector can be expressed as:

$$\begin{aligned} I_j &= k_j \cdot E_j \cdot E_j^* \\ I_y &= I_1 - I_2 \\ I_x &= I_3 - I_4 \end{aligned} \tag{9}$$

where $E_j{}^*$ is the conjugate transpose matrix of $E_j$. By calculating the Jones matrix, the parameters of the two sinusoidal intensity signals can be obtained as follows:

$$\frac{I_y}{I_x} = \frac{A + B\cos(\varphi) + C\sin(\varphi)}{D + E\cos(\varphi)} \tag{10}$$

where $\varphi(\varphi = \varphi_m - \varphi_r)$ is the phase difference between the reference beam and the measuring beam. Equations (5)–(8) are substituted into Equations (9) and (10), and thus, the parameters of Equation (10) can be written in detail as follows:

$$\begin{aligned} A &= \left[ k_2\left(r_{s2}^2 + r_{p2}^2\right) - k_1\left(t_{p2}^2 + t_{s2}^2\right) \right] \cdot t_{p1}^2 r_{s1}^2 \\ B &= 2\left[ k_2\left(r_{s2}^2 + r_{p2}^2\right) - k_1\left(t_{p2}^2 + t_{s2}^2\right) \right] \cdot t_{p1} r_{p1} t_{s1} r_{s1} \\ C &= \left[ k_2\left(r_{s2}^2 - r_{p2}^2\right) + k_1\left(t_{p2}^2 - t_{s2}^2\right) \right] \cdot t_{p1}^2 r_{s1}^2 \\ D &= \left[ k_4\left(r_{s3}^2 + r_{p3}^2\right) - k_3\left(t_{p3}^2 + t_{s3}^2\right) \right] \cdot t_{p1}^2 r_{s1}^2 + 2\left[ k_4\left(r_{s3}^2 - r_{p3}^2\right) + k_3\left(t_{p3}^2 - t_{s3}^2\right) \right] \cdot t_{p1} r_{p1} t_{s1} r_{s1} \\ E &= 2\left[ k_4\left(r_{s3}^2 + r_{p3}^2\right) - k_3\left(t_{p3}^2 + t_{s3}^2\right) \right] \cdot t_{p1} r_{p1} t_{s1} r_{s1} + \left[ k_4\left(r_{s3}^2 - r_{p3}^2\right) + k_3\left(t_{p3}^2 - t_{s3}^2\right) \right] \cdot t_{p1}^2 r_{s1}^2 \end{aligned} \tag{11}$$

Here, the values of $r_p{}^2$ and $t_s{}^2$ are extremely small, such that multiplying $r_p{}^2$ by $t_s{}^2$ can yield zero.

When the gains of the four interference signals are the same (i.e., $k_1 = k_2 = k_3 = k_4$), parameters *A*, *B* and *D* are not equal to zero in Equation (11), whereas parameter *C* is not equal to *E*. Parameters *A* and *B*, the first term of *D* and the first term of *E*, which are induced by the PBS$_2$ and the PBS$_3$ in the detection part, can be adjusted to zero and then removed by adjusting the gains of the detectors. However, the cross term of $t_{p1}r_{p1}t_{s1}r_{s1}$ that corresponds to the polarization mixing of the PBS$_1$ of the interferometer part cannot be eliminated. Therefore, polarization mixing is the primary factor that causes the nonlinear error for the PBS$_1$ in the interferometer part. For the PBS$_2$ and the PBS$_3$ in the detection part, the unequal separation of optical intensity is the dominant factor that results in the nonlinear error. Therefore, the nonlinearity caused by the imperfections of the PBSs at different positions of the homodyne interferometer should be compensated using different means. For the imperfections of the PBS$_1$ in the interferometer part, the focus is on removing the unwanted s-polarized beam in the transmitted beam and the unwanted p-polarized beam in the reflected beam. For the imperfections of the PBS$_2$ and the PBS$_3$ in the detection part, the key is to make the transmitted intensity-to-reflected intensity ratio approach 1.

The improvement of the traditional gain-adjustment method can be evaluated by assuming that *A*, *B* and the first terms of *D* and *E* are all equal to zero. The following conditions for eliminating nonlinear error can be obtained:

$$\frac{k_2}{k_1} = \frac{t_{s2}^2 + t_{p2}^2}{r_{s2}^2 + r_{p2}^2}$$
$$\frac{k_4}{k_3} = \frac{t_{s3}^2 + t_{p3}^2}{r_{s3}^2 + r_{p3}^2}$$
$$\tag{12}$$

Considering the symmetry of the optical layout of the detection part, we select the PBS$_2$ and the PBS$_3$ with the same performance parameters, thereby indicating that the reflection and transmission for all the PBSs are the same. The actual parameters of the PBSs are shown in the Table 1.

**Table 1.** The parameters of the optical components.

| Optical Component | $t_p{}^2$ | $t_s{}^2$ | $r_p{}^2$ | $r_s{}^2$ |
|---|---|---|---|---|
| PBS$_1$ | 0.95 | 0.01 | 0.05 | 0.99 |
| PBS$_2$ | 0.95 | 0.01 | 0.05 | 0.99 |
| PBS$_3$ | 0.95 | 0.01 | 0.05 | 0.99 |

Therefore, the relationships of the gains of four photodetectors can be obtained as follows:

$$k_1 = k_3 \qquad k_2 = k_4$$
$$\frac{k_2}{k_1} = \frac{k_4}{k_3} = 0.932$$
$$\tag{13}$$

By introducing the preceding modified results into Equation (11), Equation (10) can be rewritten as follows:

$$\frac{I_y}{I_x} = \frac{t_{p1}^2 r_{s1}^2 \sin\varphi}{2 \cdot t_{p1}r_{p1}t_{s1}r_{s1} + t_{p1}^2 r_{s1}^2 \cos\varphi}$$
$$\tag{14}$$

As shown in Equation (14), gain adjustment can only eliminate the nonlinear errors caused by the PBS$_2$ and the PBS$_3$ in the detection part. Suppose that the PBS$_2$ and the PBS$_3$ are ideal components and only the PBS$_1$ is not ideal. Then, the result of the two quadrature signals is the same as that of Equation (14), which shows that gain correction cannot compensate for the polarization mixing caused by the PBS$_1$ in the interferometer part.

Here, common commercial PBSs with the properties as mentioned above are chosen, and the nonlinear errors before and after gain correction are simulated using MATLAB software. The results without gain adjustment, i.e., Equation (10), and with gain adjustment, i.e., Equation (14), are introduced

into Equation (4), respectively. Then, the values of the nonlinear errors are simulated, as shown in Figure 2, where the wavelength of the laser is 632.8 nm. The blue dotted curve shows the nonlinear error without any correction, and the peak-to-peak value is approximately 10 nm. The red solid curve shows the nonlinear error when gain correction is applied, and the peak-to-peak value is approximately 4.5 nm. The preceding theoretical analysis and simulation show that reducing the nonlinear error by adjusting only the gains of the detectors for a general PBS is ineffective.

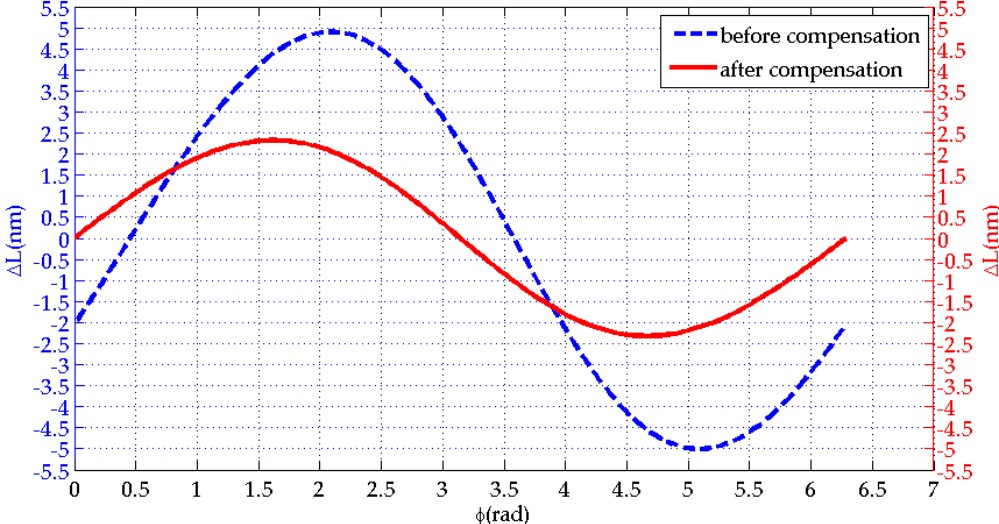

**Figure 2.** The comparison of nonlinearity before and after the gain adjustment: the blue dotted curve—before gain correction; the red solid curve—after gain adjustment.

### 3. Compensation for the Polarization Mixing of $PBS_1$

The nonlinear error caused by the PBSs in the detection part can be reduced by adjusting the gains of the four photo-detectors, but the polarization mixing caused by the PBS in the interferometer part cannot be eliminated [17]. To solve this problem, an optical compensation method is proposed to eliminate the polarization mixing of the $PBS_1$ in the interferometer part. Figure 3 shows the optical layout of the homodyne interferometer with an optical compensation part. The separation of the reference and measuring beams, which become two separate parallel beams, at the end of the interferometer part is critical. To achieve this, a right-angle prism is used to replace the planar mirror as the reference mirror. The unwanted s-polarized component in the p-polarized beam and the unwanted p-polarized component in the s-polarized beam are isolated through the $P_2$ and the $P_3$, respectively, with a high extinction ratio of approximately 10,000:1. Then, the pure polarized measuring and reference beams are combined by the $PBS_0$. The polarization mixing of the $PBS_1$ is theoretically eliminated by adopting this optical compensation. Thus, the inherent polarization mixing of the homodyne interferometer caused by the $PBS_1$ can be suppressed.

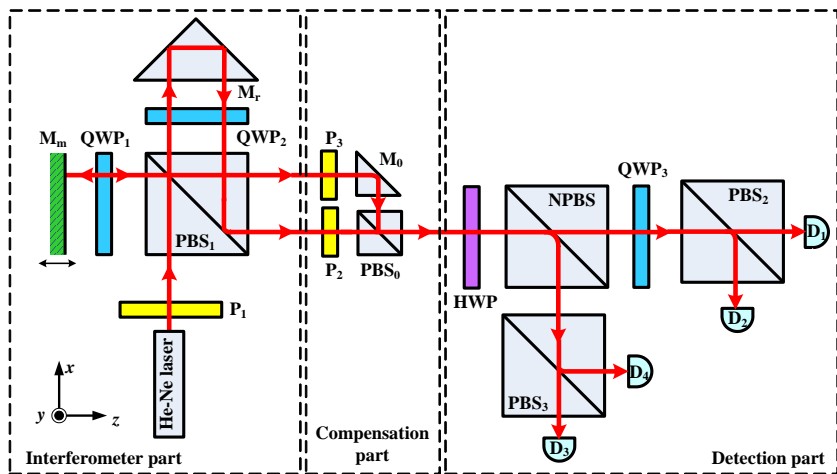

**Figure 3.** Homodyne interferometer with a compensation part. $P_1$, $P_2$ and $P_3$ are the polarizers; $PBS_0$, $PBS_1$, $PBS_2$ and $PBS_3$ are the polarizing beam splitters; $M_r$ is the mirror; $M_m$ and $M_0$ are the right-angle prisms; $QWP_1$, $QWP_2$ and $QWP_3$ are the quarter-wave plates; HWP is the half-wave plate; NPBS is the non-polarizing beam splitter; and $D_1$, $D_2$, $D_3$ and $D_4$ are the photoelectric detectors.

To prevent the generation of the unwanted s- and p-polarized components, the transmission axes of the $P_2$ and the $P_3$ should coincide with the axes of the reference and measuring beams, respectively, thereby ensuring that the measuring pure p-polarized beam and the reference pure s-polarized beam are produced. The ideal Jones matrices of the $P_2$ and the $P_3$ with a high extinction ratio can be expressed as follows:

$$P_2 = \begin{bmatrix} 0 & 0 \\ 0 & 1 \end{bmatrix} \qquad P_3 = \begin{bmatrix} 1 & 0 \\ 0 & 0 \end{bmatrix} \tag{15}$$

Assuming that the $PBS_0$ and the $M_0$ are ideal optical components is reasonable because the incident beams passing through the $PBS_0$ are nearly ideal linear polarized beams that originate from the $P_2$ and the $P_3$ with a high extinction ratio. Even if the $PBS_0$ has an imperfect property, it will only induce optical energy loss but will not cause polarization mixing. This nonlinear error can be compensated for by adjusting the gains of the detectors. The ideal matrix Equation (15) of the $P_2$ and the $P_3$ is substituted into Equation (8). The two intensity signals of the homodyne interferometer can be obtained by recalculating Equations (5)–(10) as follows:

$$\frac{I_y}{I_x} = \frac{[k_2(r_{s2}^2 + r_{p2}^2) - k_1(t_{p2}^2 + t_{s2}^2)]t_{p1}^2 r_{s1}^2 + [k_2(r_{s2}^2 - r_{p2}^2) + k_1(t_{p2}^2 - t_{s2}^2)]t_{p1}^2 r_{s1}^2 \sin\varphi}{[k_4(r_{s3}^2 + r_{p3}^2) - k_3(t_{p3}^2 + t_{s3}^2)]t_{p1}^2 r_{s1}^2 + [k_4(r_{s2}^2 - r_{p2}^2) + k_3(t_{p2}^2 - t_{s2}^2)]t_{p1}^2 r_{s1}^2 \cos\varphi} \tag{16}$$

As shown in Equation (15), the cross term of $t_{p1}t_{s1}r_{p1}r_{s1}$ caused by the polarization mixing of the $PBS_1$ has been completely eliminated after using the $P_2$ and the $P_3$ with a high extinction ratio, the $PBS_2$ and the $PBS_3$ have the same parameters as those listed in Section 2, and gain adjustment is applied according to Equation (12). From Equation (15), parameters A and D are equal to zero, whereas parameter C is equal to parameter E. At this point, the signals $I_x$ and $I_y$ have become quadrature signals in Equation (15). That is, all nonlinear errors have been theoretically eliminated.

In reality, the linear polarizers with a high extinction ratio are not perfect but experience a small amount of light leakage. Suppose that the position and angle errors of the optical components are disregarded. The Jones matrix of the $P_2$ and the $P_3$ with an extinction ratio of 10,000:1 can be rewritten as follows:

$$P_2 = \begin{bmatrix} \varepsilon & 0 \\ 0 & 1 \end{bmatrix} \qquad P_3 = \begin{bmatrix} 1 & 0 \\ 0 & \varepsilon \end{bmatrix} \tag{17}$$

where $\varepsilon$ is the extinction ratio and is equal to 0.01 (i.e., $\varepsilon^2 = 0.0001$). The extinction ratio $\varepsilon$ of the polarizer before and after the correction of the Jones matrix is shown in Table 2.

**Table 2.** The extinction ratio $\varepsilon$ before and after correction of Jones matrices.

| Polarizer | $\varepsilon$ before Correction | $\varepsilon$ after Correction |
|:---:|:---:|:---:|
| $P_1$ | 0 | 0.01 |
| $P_2$ | 0 | 0.01 |

The real matrix Equation (17) of the $P_2$ and the $P_3$ is substituted into Equation (8). Then, the two intensity signals can be obtained by adopting the gain coefficients of the four detectors according to Equation (12) as follows:

$$\frac{I_y}{I_x} = \frac{r_s^2 t_p^2 \sin(\varphi)}{2\varepsilon \cdot t_p \cdot r_s \cdot r_p \cdot t_s + r_s^2 t_p^2 \cos(\varphi)} \tag{18}$$

Equations (18) and (14) represent the results with and without the optical compensation method, respectively. To compare the nonlinear errors in the two cases, the nonlinear errors are simulated and compared when Equations (14) and (18) are substituted into Equation (4).

The comparison results are illustrated in Figure 4. The blue dotted curve shows the nonlinear error when only gain adjustment is adopted, and the peak-to-peak value is approximately 4.5 nm. The red solid curve shows the nonlinear error when high-quality polarizers are combined with gain adjustment, and the peak-to-peak value is approximately 0.045 nm. The nonlinear error is reduced by two orders of magnitude after adopting the high-quality polarizer, and polarization mixing caused by the PBS$_1$ in the interferometer part is also evidently decreased. A residual error of 0.045 nm is observed, which is attributed to the high-quality polarizer not being completely ideal. If the extinction ratio is increased further, then this error will be reduced further.

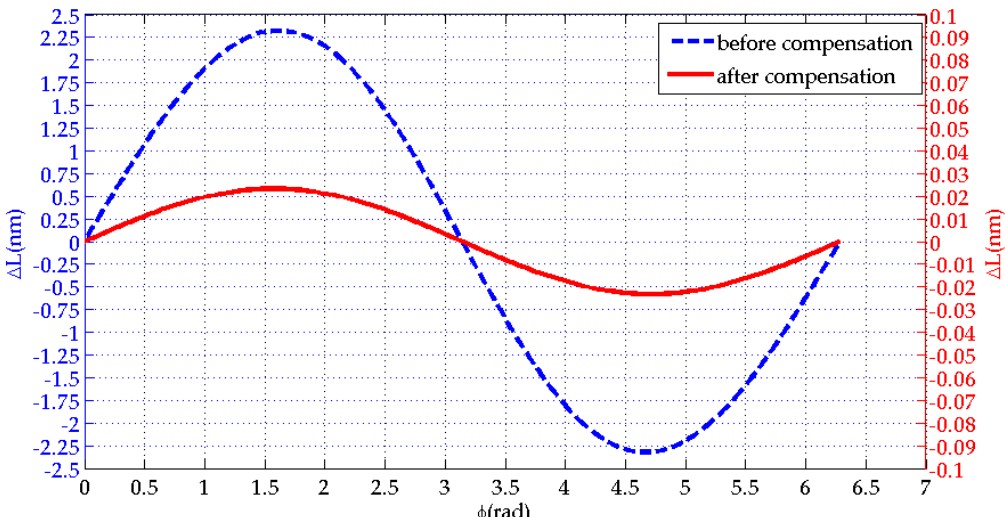

**Figure 4.** The comparison of nonlinearity before and after adoption of the high-quality polarizer: the blue dotted curve—before adoption of the high-quality polarizer; the red solid curve—after adoption of the high-quality polarizer.

## 4. Conclusions

Through the theoretical calculation and analysis of interference signals, the influence of the PBS in the interferometer part on nonlinearity is determined to be embodied in polarization mixing, whereas the effect of the PBSs in the detection part on the nonlinearity is the unequal separation of optical intensity. The DC offsets and unequal AC amplitudes caused by the PBSs in the detection

part can be easily eliminated by adjusting the gains of the detector. However, this approach does not work for the polarization mixing caused by the PBS in the interferometer part. In this regard, the nonlinear error caused by the polarization mixing of the PBS in the interferometer part is analyzed, and an optical configuration to compensate for the polarization mixing caused by the PBS in the interferometer part is proposed. By placing two high-quality polarizers at the end of the interferometer part, the unwanted s- and p-polarized components are isolated by the polarizers, thereby suppressing the inherent polarization mixing of the homodyne interferometer. The nonlinearity is reduced from approximately 4.5 nm to 0.045 nm, which is a decrease of two orders of magnitude, by integrating other compensation methods such as gain correction, compared with using only gain adjustment. The residual error is mainly due to the imperfection of the polarizer.

Compared with the existing studies, we focus on the compensation of the polarization mixing of imperfection of the PBS, which is difficult to correct. In our proposed solution, polarization mixing caused by imperfection of the PBS is isolated by the polarizers with a high extinction ratio in the optical layout; thereby, the inherent nonlinear error of the homodyne interferometer is reduced. In a future experiment, all of the optical components in the optical layout should be considered. DC offsets and unequal AC amplitudes that are induced by the other optical components except the PBS can be reduced first through fine adjustment and improved components. Then, the nonlinear errors can be further corrected by our proposed optical layout of compensation and combination with gain adjustment.

**Author Contributions:** All work in relation to this paper was accomplished by all authors' efforts. Conceptualization, Q.H. and C.W.; methodology, C.W.; simulation and visualization, X.D., R.C., L.Z., R.L. and H.L.; writing the paper, C.W. All authors have read and agreed to the published version of the manuscript.

**Funding:** This research was funded by National Natural Science Foundation of China (Grant No. 51875163, 51705123), National Key Research and Development Program of China (Grant No. 2019YFB2004900) and Anhui Provincial Natural Science Foundation (Grant No. 1908085QE201).

**Acknowledgments:** The authors acknowledge the financial support from National Natural Science Foundation of China (Grant No. 51875163 and 51705123), National Key Research and Development Program of China (Grant No. 2019YFB2004900) and Anhui Provincial Natural Science Foundation (Grant No. 1908085QE201).

**Conflicts of Interest:** The authors declare no conflict of interest.

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
