# Peer review of "Compensation Method for Polarization Mixing in the Homodyne Interferometer"

_applsci, doi:10.3390/app10176060_

Round 1
Reviewer 1 Report
This is a strictly theoretical paper which makes a lot of assumptions about many of the optical components in the systems described. Every component not only has a manufacturing error but also a setup error associated with their angular positioning. Every polarizer has leakage in the non-axis direction, the transmission phase shift of the waveplates also has a variation which is adjusted usually by a rotation about its axis. These components also must be set along particular horizontal angle. These all provide bias as well as uncertainty in the effective phase shifts. The manufacturer specs provide an uncertainty and not a particular difference that is known a priori. Each section of the component must be considered from the very first polarizer after the laser source. The addition of the additional components also further compounds these errors. There is also a lack of experimental data to verify the operation of the nonlinearity reduction.
There are also some mathematics problems in the reported Jones matrices which will affect your results. There is a factor of 0.5 missing in the reflection and transmission matrices. A more important error is in the equation for the transmission through a half-wave plate which does not shift the phase of one of the components via a multiplication by a complex number. At what angle of orientation can all the components of H have the same imaginary magnitude?
The general angle dependent Jones equations for the components should be introduced first and show how there misalignment will influence the output as the path difference in the interferometer changes.
Ideally an experimental portion would accompany the theoretical as confirmation of results. Otherwise every detail must be considered in manufacturing errors and alignments.
Also since actual parameters of the optics are unknown a method for determining the adjustments which need to be made to achieve the maximum reduction of linearity needs to be considered.
Reviewer 2 Report
The authors present a fully numerical demonstration of a new compensation method for polarization mixing in homodyne interferometers that can be of interest for readers of applied sciences focusing on this line of work, but the manuscript itself presents, in my opinion, a series of problems that should be addressed before it can be considered acceptable for publication:
1 - The paper closely follows the structure of that of Ahn et al., Optics Express 17 (25), pp. 23299-23308, down to the analytical approach and the limited number of references. The authors should improve the provided background on the problem and expand on the references, including both some on the relevance of the problem at hand and others on different proposed solutions, that they should later compare with their proposed one.
2 - The authors should include a table with the parameters used in the simulation, considering their potential variability, instead of saying simply that they used realistic ones.
3 - They should also adequately compare their solution to other proposed methods in the literature and discuss when and why their proposed approach is relevant and useful. Just mentioning an improvement of two orders of magnitude out of context is not enough, particularly in a fully numerical paper.
After such changes are implemented, I would be willing to re-review the manuscript.
Round 2
Reviewer 1 Report
The Jones equations presentation is much better.
One grammatical error:
"with consisted of two" should be which consisted of...
Looking forward to the experimental results.
Reviewer 2 Report
I thank the authors for their effort. I think they have improved the paper considerably, so it can now be accepted for publication, after some additional English language revision, particularly over the new sections.
